# Translating School Faculty Experiences Using PBIS into Recommendations for Practice

**DOI:** 10.3390/bs13050372

**Published:** 2023-05-02

**Authors:** Hannah M. Terrell, Su-Je Cho

**Affiliations:** Graduate School of Education, Fordham University, 113 West 60th Street, New York, NY 10023, USA

**Keywords:** positive behavior interventions and supports, PBIS, special education, school faculty perceptions, program satisfaction, professional development, administrative support

## Abstract

Positive Behavior Interventions and Supports (PBIS) is a behaviorally based framework that seeks to improve student outcomes in schools. This framework is implemented at differing levels of intensity within a school based on students’ unique needs. Special education teachers and school psychologists are integral pieces of PBIS implementation. Within the context of the COVID-19 pandemic, these service providers may face unique challenges in implementing PBIS principles in schools, particularly due to new or adapted role demands and increased feelings of burnout. The current study examined special education teachers’ and school psychologists’ perceptions of their schools’ practices related to PBIS in the wake of the COVID-19 pandemic within five dimensions of understanding and school-based support, as well as overall satisfaction with PBIS in their school. Opportunities for professional development and the presence of PBIS teams emerged as major contributors to faculty satisfaction; however, only about half of participants indicated access to these resources. Special education teachers indicated higher levels of satisfaction with their administrative support and school communication practices when compared to school psychologists. Best practices and reflections from interview participants are discussed.

## 1. Introduction

Positive Behavior Interventions and Supports (PBIS) is an evidence-based, tiered framework supporting students for improved outcomes in academic, behavior, emotional, and social development [1]. PBIS combines a behavioral approach with multi-tiered support to address student needs and has been implemented in more than 21,000 schools throughout the United States [2,3]. PBIS programs have been implemented in a variety of schools, as schools are allowed to create their own behavioral goals, and reward system creation is left up to individual schools based on their cultural context and needs [2].

PBIS has been utilized to manage externalizing behaviors, such as bullying, fighting, and class disruptions [4]. Randomized control trials within elementary schools demonstrated the effectiveness of PBIS in increasing school safety perceptions and increasing academic assessment scores in reading, as well as attesting to the positive effects of professional development for schoolwide implementation [5]. Longitudinal studies in middle and high schools using PBIS have found a positive correlation between the number of years a PBIS program has been implemented and positive school outcomes, such as fewer office discipline referrals, higher fidelity of implementation, and more positive student perceptions of school climate [6]. The PBIS approach can also benefit students who exhibit internalizing problems, such as anxiety, withdrawal, sadness, and somatization [4]. Cook and colleagues [4] found that classrooms implementing PBIS showed greater decreases in externalizing behaviors when compared to those without any prevention-based program [4]. Moreover, combining the PBIS framework with additional social–emotional learning (SEL) modules has demonstrated further improved outcomes by reducing both internalizing and externalizing behaviors compared to those in classrooms [4]. At the Tier 2 level, the PBIS framework has been shown to be effective in reducing bullying incidents, as well as reducing both victims’ and bystanders’ socially reinforcing responses that may have previously maintained bullying behaviors [7].

### 1.1. What Does Successful PBIS Implementation Look Like?

The ongoing success of PBIS in a school is evaluated based on how closely the school follows specified guidelines set forth by the National Center on PBIS. Schools that have these procedures in place to the expected degree are implied to have high fidelity in their program implementation. The Schoolwide Evaluation Tool (SET) is commonly used to evaluate fidelity of implementation annually [8]. Metrics of implementation fidelity are used in the SET to evaluate expected involvement of staff in setting expectations for behavior and dissemination of behavioral expectations among staff and students [8]. In addition, implementation of PBIS with fidelity relies on thorough data-based decision-making procedures related to student behaviors and discipline referrals led by a dedicated school PBIS team [8].

Evaluation methods such as the SET provide structure and support for long-term PBIS success and achievement of positive student outcomes, including fewer suspensions, higher academic achievement, and lower rates of truancy [2,9]. PBIS implementation with fidelity has also demonstrated a positive impact on student outcomes over time, including greater decreases in out-of-school suspensions (OSSs) the longer a school has used PBIS with fidelity [10]. The PBIS framework relies heavily on school stakeholders to implement procedures and use data-based decision making to reach students effectively. This reliance places a large responsibility for the program’s efficacy on the shoulders of school administrators, teachers, and other personnel who are involved in schoolwide programming, including school psychologists. School personnel are responsible for implementing PBIS, in addition to instruction and curriculum development, parent and community involvement, lesson planning, and other necessities for the school to function. Thus, it is critical to ensure that school personnel have the necessary knowledge and resources to implement PBIS with fidelity while balancing daily responsibilities. To further ensure faculty’s needs are being met, PBIS procedures should be routinely evaluated.

### 1.2. The Role of Administrative Support

School administrators play an important role in successful implementation of PBIS with fidelity in their schools. The provision of support and buy-in from administrators affects how efficacious and bought-in other school stakeholders can be. Administrators can provide differential support from the initial exploration of PBIS in the school through the maintenance phase [11]. For example, administrators set the tone by establishing schoolwide goals, providing necessary resources, and modeling expected behaviors for students [11]. In addition, administrators can monitor the progress of PBIS implementation through data-based decision-making procedures [11]. It is crucial that administrators not only provide and maintain PBIS databases in the school, but also make this data available to staff members to sustain the program over time [11].

McDaniel et al. [12] noted that lack of support or changes in administration can disrupt PBIS implementation. Furthermore, the lack of understanding and buy-in and the focus on punitive measures of discipline can harm positive student outcomes and climate within the school. Thus, fostering buy-in and climate requires consistent communication to understand school stakeholders’ perceptions of how the program is working, as well as regular sharing of progress and modification of goals related to PBIS data [11].

### 1.3. PBIS Communication and Logistics

Administrator communication with staff and students is a crucial component of PBIS success. Administrators set expectations for how staff will implement PBIS in classrooms and other areas of the school, such as the playground, cafeteria, and hallways [11]. Because of this, it is crucial that administrators create opportunities to communicate with school stakeholders regarding PBIS implementation [13]. One avenue of this communication is to not only attend, but also disseminate information from PBIS team meetings to all staff members as a means of keeping the school community informed [11,13]. This information sharing should also impact students, as providing clear definitions and expectations for behavior is a component of implementation fidelity [8,14]. McKevitt and Braaksma [15] suggest that at least one administrator needs to be heavily involved in PBIS, from modeling expected behaviors to attending all meetings and investing both time and money into program success.

Administrators also play an important role in overseeing the school’s PBIS team. For example, they can select effective members who represent the school staff, provide resources for the team to be successful, and attend the team’s meetings on a regular basis [11]. Having a dedicated PBIS team is also a requirement for schools to comply with expected implementation fidelity procedures [14], and administrators are expected to participate within this team [8].

### 1.4. The Importance of Professional Development

Staff professional development (PD) related to PBIS is critical for the framework to be successful. As such, PD is a component of implementation fidelity that schools are expected to provide [14]. This development may first take place at the district or state level and often involves training the school’s dedicated PBIS team prior to development for all staff [15]. Effective staff development may take multiple years, typically offered either at the beginning of the school year or in small increments throughout the year, and may become the responsibility of the school’s dedicated PBIS team to initiate [15]. PD is key to ensuring that staff understand implementation expectations, that they are knowledgeable about the framework and three tiers of prevention, and that ongoing issues or concerns can be addressed in a learning environment [15]. This is especially important for new staff entering the school for the first time [12].

PBIS-related PD can be distinctly helpful for school staff who feel unprepared when behavioral challenges arise in the classroom. McDaniel et al. [12] described how the complexity of these challenges can feel overwhelming for classroom teachers, as they may not have prerequisite knowledge about how to carry out positive interventions and may resort to “easier” punitive strategies that result in power struggles between teachers and students. Providing a solid foundation of knowledge in PBIS theory and practice for teachers and other school staff can aid in fidelity of implementation that benefits both the student population and teachers, who may otherwise feel overwhelmed.

### 1.5. Use of PBIS Amidst the COVID-19 Pandemic

In the wake of the COVID-19 pandemic, school professionals have been expected to wear several hats and willingly take on increasing amounts of work to bridge gaps in student learning, as it has become more apparent than ever how integral school services are to community functioning. Around the world, school psychologists’ consultation, counseling, and to a lesser extent, assessment services shifted online, drastically changing their daily landscape and bringing about new technological demands for practitioners [16]. In the United States, school psychologists reported greater time spent practicing consultation and seeking effective interventions for teachers during the pandemic compared to previous service provision [16]. School psychologists also reported providing online resources for families related to behavioral and social–emotional concerns [16]. Similarly, daily tasks and responsibilities for special education teachers shifted extensively during the pandemic. Special education teachers faced unique pressure to continue providing instruction that met students’ unique needs as specified in IEPs or 504 plans and tended to spend more time both working with children and families and preparing their remote instruction methods than general education teachers [17]. Special education teachers also reported spending more time focusing on progress-monitoring tasks and checking in on the well-being of students with behavioral or emotional disorders compared to other related service providers, including school psychologists, during the pandemic [18]. Both special education teachers and school psychologists, in addition to other school-based related service providers, frequently used online platforms such as Zoom [18].

In January 2022, the National Education Association asked over 3500 of their members to rate their perceived seriousness of current issues in education [19]. Of these issues, the respondents rated “feeling burned out” as the most serious, with 66% stating it was a “very serious” issue and 90% rating it “serious” overall [19]. In addition, 76% rated behavioral issues in students as a “serious” issue, with 44% of the total respondents going so far as to call it a “very serious” issue [19]. At this unique time in history, school personnel burnout combined with serious behavioral concerns with students could pose unique challenges to PBIS implementation. On one hand, heightened concerns about student behaviors following the return to in-person instruction suggests an increased need for structured behavioral support in the classroom. Here, PBIS can play an important role in providing clear and consistent guidelines for students to follow and systematized rewards that can make space for more positive student outcomes. On the other hand, increases in burnout among school personnel amidst the pandemic pose a risk to the successful implementation of PBIS programs in schools. Having just experienced an extended period during which much school programming was adapted on the fly, intermittent, or abandoned altogether due to sudden distance education requirements, it is to be expected that school personnel feel burned out or are struggling to resume “business as usual” operations. Notably, teacher self-efficacy in their behavior management skills was found to significantly predict their own well-being just after the onset of the pandemic [20], suggesting a clear link between school personnel well-being and behavioral management—the two primary concerns noted by educators in January 2022 [19].

In March 2020, the Center on PBIS [21] released initial guidance for school personnel regarding how they may continue to implement basic PBIS values during the onset of the COVID-19 pandemic. This preliminary guidance focused on how typical behavioral expectations could be extended to health necessities, such as handwashing or use of cleaning products [21]. In addition, the Center on PBIS [21] suggested highlighting community support, including use of behavioral expectations to protect against racialized harassment and extending guidance to families on how behavioral expectations may be used at home. This guidance remained generalized in nature, likely due to the rapidly changing circumstances of the time, and as it stands, little to no guidance on use of PBIS amidst online or distance education was available at the onset of the pandemic.

Following this initial provision, more detailed guidance emerged in the summer of 2020 as the Center on PBIS collaborated with several other educational organizations [22]. This document provided guidance for school and district personnel in the aftermath of a crisis more broadly, not only in the wake of COVID-19 [22]. However, this document was able to outline considerations and examples for school personnel that encompassed online learning environments and public health concerns. Selected examples of guidance for remote learning implementation include provision of modified curriculum and interventions by a school leadership team, adapting behavioral expectations within distance education, use of virtual methods for progress monitoring, and use of multimodal engagement strategies during instruction [22]. Notably, no direct suggestions were made for hybrid learning modalities, as schools were instructed to select strategies from both remote and in-person techniques as they saw fit [22].

### 1.6. How Fidelity of Implementation Affects School Personnel

Although improving student outcomes is an overarching goal of PBIS, the effects of positive school climate and strong PBIS implementation may extend beyond students alone. In comparing teacher ratings of stress and coping just before and just after the onset of the pandemic, Herman et al. [20] found that both mutually respectful leadership and perceptions of equitable discipline practices in the school were predictive of teacher well-being. This provides a foundation for PBIS implementation to support school faculty well-being, as both positive leadership skills from a principal and just discipline practices are two cornerstones of the PBIS framework.

With a hypothesis that positive school climate can play a role in mitigating teacher turnover, Ross et al. [23] examined the well-being, efficacy, and burnout of teachers in elementary schools that had high SET scores compared with their counterparts with low SET scores for their PBIS programs [23]. The results showed that high fidelity of implementation, indicated by higher SET scores, was significantly correlated with higher feelings of teacher efficacy and personal sense of accomplishment, as well as lower feelings of emotional exhaustion in comparison with teachers at schools with low implementation fidelity [23]. This suggests that the well-being of school stakeholders that are implementing PBIS programs is a key ingredient in program success and positive student outcomes. Similarly, school socioeconomic status (SES) moderated these relationships in that the effects of PBIS implementation fidelity were most strongly observed in schools with low SES when compared to schools with high SES [23]. This result indicates that PBIS implementation with high fidelity can be a tool for positive outcomes for teachers in schools with limited resources [23], ideally reducing teacher burnout and turnover in these areas.

Beyond burnout and efficacy, additional factors may work against school personnel’s ability to implement PBIS with fidelity. Lawson et al. [24] interviewed 17 teachers at PBIS schools to examine factors that affected their use of behaviorally based interventions on children with ADHD. Several factors that hindered their successful use of behavioral interventions in the classroom were time constraints due to academic curriculum needs and unexpected interruptions, expectations of internal motivation for positive behavior particularly as students get older, and the need for repeated practice to feel efficacious [24]. As aids for their use of behavior intervention, the respondents listed administrative support, building rapport with students through knowledge of their individual style and needs, visual cues and reminders related to behavioral interventions, and advance learning opportunities provided by the school, such as ongoing trainings and provision of real-life examples [24]. Factors which aid in the use of classroom behavioral interventions have been found to largely rely on a teacher’s understanding of the efficacy and significance of the given intervention strategy [24]. This suggests that buy-in from school stakeholders is a critical element that affects whether behavioral strategies will be utilized effectively in a classroom.

It is uniquely important to examine the perceptions of special education teachers and school psychologists as they relate to PBIS implementation. Research has suggested that special education teachers may be more confident in their PBIS usage compared to other school staff, which in turn relates to greater sustainability of PBIS implementation in their school [25]. This finding acknowledges special education teachers as a particularly valuable resource within the overall success of PBIS models in school and makes the provision of resources and support to them a high priority for administrators. As such, school psychologists play an important consultative role at the systems level within a school, using their PBIS training to influence program success by serving on PBIS teams, assessing implementation fidelity, and providing guidance for data-based decision-making practices [15,26,27]. However, school psychologists engaged in PBIS work have reported difficulty navigating varied levels of buy-in from staff and school administrators and deficits in communication [27], which may ultimately affect their effectiveness in program delivery with fidelity.

### 1.7. Current Study

The current study aimed to investigate special education teachers’ and school psychologists’ experiences using the PBIS framework in their schools to inform best practices suitable for field use. The authors examined two major research questions: (1) In the context of COVID-19, how satisfied are special education teachers and school psychologists with their schools’ PBIS programs? (2) What, if any, differences in PBIS satisfaction exist based on type of role (i.e., special education teacher or school psychologist), the presence of a specified PBIS team, the use of PD, or level of participant experience in their role and with PBIS in general?

## 2. Method

All data were collected between November 2021 and November 2022, primarily during the winter and early spring of 2022. An explanatory sequential mixed-methods design was used [28], whereby the results of an initial quantitative segment are further elucidated in a secondary qualitative segment. ANOVA F tests were used to determine group mean differences related to the independent variables in this study.

In the qualitative portion, interviewees (seven special education teachers and one school psychologist) provided meaningful responses when asked to elaborate on topics presented in the survey they previously engaged in (see “Measures” below), which allowed the researchers to better understand participant experiences related to PBIS and make inferences regarding the importance of the topics discussed. In addition, interviewees gave insight into best practices they had engaged in based on the selected topics, such as ways in which their school communicated information about PBIS rewards and procedures. Analyses of qualitative interviews were conducted using a thematic content analysis method based in phenomenological principles, whereby interview data are gathered and objectively described and grouped based on content into themes [29,30]. These themes represent the concepts and ideas put forth by the interviewees, which are then represented by coding structures so as to make meaning from the data across participants [30]. As the purpose of this study was to seek out and explore faculty experiences using PBIS without preconceived judgments, this inductive approach was used to best capture and interpret faculty perceptions objectively [29,30]. In addition, these perceptions were analyzed and transformed into a practical understanding of faculty satisfaction and recommendations to increase satisfaction in tandem with knowledge of PBIS frameworks.

### 2.1. Participants

#### 2.1.1. Survey Participants

Detailed information about participant demographics is presented in Table 1. A total of 104 special education teachers and school psychologists responded to the initial survey. Due to misalignment with inclusion criteria or failure to complete a portion of the survey that provides meaningful data, 17 of these participants were excluded from data analyses. The remaining 85 participants were included in data analyses.

Of this larger group, 58 participants (66.7%) were special education teachers, and 29 participants (33.3%) were school psychologists. The special education teachers had worked in their current roles for an average of 3–5 years and had experience working at a PBIS school for an average of 4 years. The school psychologists had been in their current role for an average of 6–10 years and had experience working at a PBIS school for 3 years on average.

#### 2.1.2. Interview Participants

Of the 85 survey participants, 36 opted in to be contacted regarding a follow-up interview. Each of these 36 participants was contacted via email up to 3 times on a rolling basis throughout the study about the follow-up interview, and a total of 8 participants agreed to complete the interview portion, with the remaining participants failing to respond to any of the 3 email prompts. These interview participants consisted of seven special education teachers and one school psychologist, as these were the school personnel who volunteered their time to participate. Their mean age was 45.9 years (*SD* = 11.7). All participants identified as female and most identified as White/Caucasian, with one participant identifying as Asian. The interviewees practiced largely in the tri-state area of New York, New Jersey, and Connecticut. On average, these participants had worked in their current role for 6–10 years and had 4 years of experience using the PBIS program in their school.

### 2.2. Measures

A survey, developed for this study using Qualtrics, consisted of screening questions, demographic questions, and items related to participants’ experience of PBIS implementation. Two screening questions were to determine whether the participant met inclusion criteria (i.e., working as a special education teacher or school psychologist in a school that utilizes the PBIS framework). The demographic questions asked participants about their race/ethnicity, gender, and age, the type of school the participant worked in (i.e., public, private, or charter), their level of experience (e.g., highest level of education, number of years in the profession, and number of years at a school that uses PBIS), what U.S. state they practiced in, and what grade levels between preschool and twelfth grade they served.

The survey regarding PBIS implementation (see Appendix A) consists of 24 items related to 5 broad category blocks: administrative support for PBIS implementation (6 items, ω = 0.94; all omega values based on current study sample), in-school communication practices related to PBIS (4 items given to all participants (ω = 0.93), 2 criterion items, 4 items given only to PBIS team members (ω = 0.87), 4 items given only to PBIS team non-members (ω = 0.96)), understanding of PBIS logistics (4 items, ω = 0.90), PD opportunities to learn about PBIS (1 criterion item to determine eligibility, 2 items), and use of PBIS in their school amidst the COVID-19 pandemic (1 item). Each item is designed for participants to rate their level of agreement with given prompts based on their experiences at their current school on a Likert scale of 1–7 (1 = strong disagreement and 7 = strong agreement). Block presentation order is randomized across participants to avoid potential order effects, whereby it can be more certain that participants’ responses were not impacted by the order of questions. Following completion of the five category blocks, participants are asked to rate their overall satisfaction on 3 items (ω = 0.89) related to their personal PBIS implementation in their practice, whether they agree that their PBIS implementation has a positive impact on their students, and whether they feel they have enough knowledge and skills to implement PBIS effectively. At the conclusion of these overall ratings, participants are given the opportunity to opt in to a follow-up interview portion with the primary researcher by providing their email address. No other identifying information is requested from participants.

During the interview portion, participants were asked open-ended questions that reflected the content of the survey materials (i.e., “Tell me what administrative support for PBIS looks like at your school”). They were asked to elaborate on situations in which a specific practice or policy (i.e., administrative action, communication from PBIS team) was particularly helpful to them or made a significant positive difference in how PBIS is implemented in their schools and give specific examples. In addition, interviewees were asked to describe how conflicts or concerns around PBIS implementation were successfully resolved and make recommendations to other staff experiencing similar challenges based on their experiences. At the close of the interview, participants were asked to impart their wisdom, best practices, and advice based on their experiences to be shared with administrators and fellow faculty members. These interviews were audio recorded for transcription and later de-identification. Following thematic saturation, interviews were discontinued.

### 2.3. Procedures

All procedures received approval from the Institutional Review Board of Fordham University. Special education teachers and school psychologists who worked in a school that utilized the PBIS framework for at least a portion of one school year were invited to participate in this study. These participants could serve students between preschool and 12th grade. Participants were recruited through professional channels (e.g., departmental research recruitment newsletter), personal connections (e.g., reaching out to family, other students, and faculty members to share the study details with potential participants using a Snowball method), and utilizing online school directories to contact potential participants via email.

Participants were invited to take part in a 10–15 min Qualtrics survey and were informed that upon completion of the survey they would be entered into a drawing to win one of two USD 25 gift cards to Barnes & Noble. Participants were also informed that they could opt in to be contacted about a follow-up qualitative interview in which their participation would be rewarded with a USD 20 Amazon gift card. Out of a total 1582 potential participants directly contacted by the researcher, 104 (6% response rate) responded to the survey. It must be noted that not all potential participants who were contacted may work in a school that utilizes PBIS.

Participants who completed the interview portion met with the primary researcher via the Zoom platform. The interview portion provided an opportunity to expand upon the five broad topics contained in the survey in more detail. On average, these interviews lasted 30 min (*SD* = 7.3 min). Interviews opened with an invitation for the participant to share how they generally use PBIS daily in their school. This was followed by an open-ended discussion of each of the five topics, beginning broadly (i.e., “Tell me what the administrative support you receive to implement PBIS looks like on a daily basis.”) and expanding on the events and procedures mentioned by the participant.

For each thematic block, summary scores were created for each participant by taking the mean of their responses to the items presented. Items related to the presence/absence of a PBIS team and participants’ interactions with said PBIS teams are reflected in their respective summary scores and were not included in the summary of communication practices. Two items were reverse coded (“I feel overwhelmed thinking about my responsibilities implementing PBIS” in logistics and “I would need more opportunities for professional development to feel confident implementing PBIS with fidelity” in PD) prior to summary score creation. Because the subscale regarding PD opportunities to learn about PBIS demonstrated low reliability (α = 0.30), quantitative analyses for this subscale have been omitted.

Unless otherwise designated, all ANOVA tests met the statistical requirements of homogeneity of variance and normal distribution of variables. Post hoc Gabriel’s tests were used to determine the contributions of each variable to significant results. Tests that did not meet the requirement of normal distribution of variables underwent a nonparametric Kruskal–Wallis *H* test. Tests with three or more groups that did not meet the assumption of homogeneity of variance underwent Welch’s *F* test, with follow up analysis completed using a Games–Howell test, and tests with two groups that violated this assumption underwent Mann–Whitney *U* tests. These tests were selected to uncover any group-based mean differences in scores that were present in the data and ensure that proper statistical assumptions were accounted for in undertaking each analysis.

The total years of experience a participant had in their role as either a special education teacher or school psychologist were categorized as follows: beginning experience (2 years or less in role), moderate experience (3–5 years in role), substantial experience (6–15 years in role), and advanced experience (16 or more years in role). The total years of experience a participant had at a school that implements a PBIS framework were categorized as follows: beginning experience (2 years or less), moderate experience (3–5 years), substantial experience (6–10 years), and advanced experience (11 or more years). As the total years of experience in their role and the total years of experience using PBIS were not significant predictors of satisfaction in most categories, a detailed discussion of results in this area is included only when they were significant.

Thematic analysis was undertaken to analyze interview data. First, a priori coding was used by the primary researcher to create six overall themes, including administrative support, communication practices, PBIS logistics, PD, PBIS during COVID-19, and overall implementation [31]. Following this, an inductive emergent coding approach [32,33] was undertaken to allow the data to reveal prevalent themes within each category and to focus on the stated opinions of the participants. An uninvolved peer debriefing [31] was used to ensure validity of themes and codes created by the primary researcher.

## 3. Results

The authors aimed to address two research questions: (1) In the context of COVID-19, how satisfied are special education teachers and school psychologists with their schools’ PBIS programs? (2) What, if any, differences in PBIS satisfaction exist based on type of role (i.e., special education teacher or school psychologist), the presence of a specified PBIS team, the use of PD, or level of participant experience in their role and with PBIS in general?

### 3.1. Administrative Support

When examining overall administrative support received to implement PBIS, participants showed neutral to some agreement (*M* = 4.76, *SD* = 1.46). Within the category of administrative support, sixteen codes were initially identified in the interviews and then categorized into four primary sub-themes: administrators setting the tone of implementation, noticeable support from administrators, lack of administrative support, and personal characteristics.

Table 2 shows survey results and differences between special education teachers’ and school psychologists’ responses. Special education teachers were significantly more satisfied with their overall administrative support than school psychologists, with *F*(1, 81) = 8.59, *p* = 0.004, ηp^2^ = 0.10. In the interviews, the school psychologist participant described several ways in which her administration sets the tone of PBIS implementation effectively, including fostering positive culture and use of growth mindset. Other participants also identified these areas as important methods of administrative support, in addition to tracking teachers’ frequency of rewarding students, getting teachers “bought in” to program values and strategies, building positive relationships with students, personal buy-in and endorsement of program values, and consistency in enforcement of behavioral expectations.

Participants identified ways in which their administrators were not supportive, including insufficient data-based decision-making procedures, neglecting to attend meetings or district-wide trainings, and lack of buy-in to program values. Participant 5′s quote illustrates one example of this: “That principal never looked at the trends. She never looked at Friday afternoon, you know, the behaviors were escalating. What do we do about Friday afternoon?” Other personal characteristics were identified as factors that can promote or hinder PBIS success, including drive to innovate, sincere communication style, and long-term experience. Participant 5 went on to discuss how some personal administrator characteristics may be unpopular, but necessary: “She’s just all about systems. Huge micromanager, which people think is a negative, I think is a positive”.

Table 3 shows survey results and differences based on whether participants received PBIS-related professional development from their schools. Participants who received PD from their schools were significantly more satisfied with their administrative support than those who were either unsure or had not received PD from their schools, with *F*(2, 80) = 13.07, *p* < 0.001, ηp^2^ = 0.25. As discussed later, participants shared how administrators can play an important role in continuing PD opportunities throughout the school year.

Table 4 shows survey results and differences based on whether participants’ schools had dedicated PBIS teams. Participants differed in their satisfaction with administrative support based on whether their school had a PBIS team, with *F*(2, 81) = 3.40, *p* = 0.04, ηp^2^ = 0.08. Participant 3 described how her administration supports her PBIS team when they host school-wide events: “The first thing that happens is our principal will bring in subs or she’ll kind of move teachers around. So the committee members’ positions are covered”.

Level of experience in one’s role and level of experience using PBIS were not associated with differences in ratings for overall administrative support. Overall, interviewees described the importance of noticeable support from their administrators as a key component of both faculty and student buy-in. For example, participants shared how administrators may provide or secure funding for student rewards. Administrators can directly reinforce students in addition to modeling, teaching, and enforcing behavioral expectations. They may also give teachers autonomy within their class implementation, or plan whole-school student rewards. Participant 5 described how her principal used visual support to aid buy-in while providing reinforcement in the classroom: “She would do it very quietly, but when she walked in the room with her little tote bag or her little bucket, because she did her job well, the kids wanted to be on her list”.

Overall, administrative support is critical to the success of PBIS programs in schools. School faculty perceive differing levels of administrative support based on whether their schools have provided them with opportunities for PBIS-related professional development, whether their school has a team dedicated to PBIS implementation, and whether their role is that of a special education teacher or a school psychologist. The years of experience one has either in their role generally or using PBIS do not appear to moderate their perceived administrative support.

### 3.2. Communication Practices

Participants were largely neutral towards their schools’ PBIS communication practices (*M* = 4.27 *SD* = 1.67). Within the category of communication practices, nineteen codes were initially identified and then categorized into four themes: use of feedback, positively framed language, recognition of student achievements, and communicating expectations to students.

Table 3 shows survey results and differences based on whether participants received PBIS-related professional development from their schools. Participants who received PD from their schools were significantly more satisfied with their schools’ communication practices than those who were either unsure or had not received PD from their schools, with *F*(2, 82) = 12.63, *p* < 0.001, ηp^2^ = 0.24. Interviewees described how important PD is for learning how to frame behavior management positively, and here mentioned how positive language is key to student buy-in. This positive language included opportunistic framing, specific praise for effort, and consistent language use throughout the school. Participant 4 summarized what she learned about positive language in her school’s PD:

“It’s really how you respond to them, is how you can make them motivated and want to, you know, comply with it. So mentioning like, Hey, that was a great job, you know, I really like how you did this. Highlight what you like, what they did”.

Participants discussed how important it is to communicate expectations to students. They mentioned various ways their schools do this, including weekly skill focus, use of wall art and graphs, visuals based on location within the school, teaching expectations prior to giving reinforcement, and re-teaching expectations after evaluating student needs. Participant 2 described what this effort looks like in her school:

“We really bring in, we hone in on a specific skill that’s been difficult for students. And we talk about it in Monday messages…school wide, everybody shows it at the same time during the day, and then we open it up for discussions and then move on to their instruction time”.

Participants also expanded on how these avenues of communication can be used to recognize student achievements and reward students for positive behaviors. These methods included announcing students’ names over the intercom, writing students’ names on a wall mural or graph, and sending announcements of recognition to parents online.

Table 4 shows survey results and differences based on whether participants’ schools had dedicated PBIS teams. The presence or absence of a PBIS team also predicted participants’ summary scores related to communication, with *F*(2, 83) = 3.31, *p* = 0.04, ηp^2^ = 0.08. Of the 87 participants, 45 identified that their school had a dedicated PBIS team (51.7%). These participants responded to four items related to PBIS team communication and support. In total, 17 participants stated they were members of the PBIS team (37.8% of those with a team, 19.5% of overall sample). Generally, participants who were members of their schools’ PBIS teams agreed that their communication and support practices were consistent and effective (*M* = 5.81, *SD* = 0.87). Participants whose schools have PBIS teams but who are not part of them showed mild agreement with items related to PBIS team support and communication (*M* = 4.90, *SD* = 1.50). Interviewees described how PBIS teams seeking frequent feedback on rewards can help to bolster student engagement. Participant 7 elaborated on how her school’s PBIS team seeks feedback:

“We also get…generated forms through Microsoft Teams where we have to fill out, okay. How did the students like the levels they earned? Did the students like what we had this time? Like really a feedback, so that the things keep changing and the students are more encouraged to earn what they like”.

Level of experience in one’s role and level of experience using PBIS were not associated with differences in ratings for overall communication practices. Collaboration between new and experienced teachers emerged in the interviews as one helpful avenue of PBIS communication, as described by Participant 1: “I share a room with someone who’s just starting out, who just graduated her undergrad. And it’s such a fresh perspective because we literally have guided one another”. This sentiment was echoed by the school psychologist interviewee regarding classroom teachers. However, special education teachers showed significantly higher levels of satisfaction with their schools’ communication practices compared to school psychologists, with *F*(1, 83) = 8.18, *p* = 0.005, ηp^2^ = 0.09. Table 2 shows survey results and differences between special education teachers’ and school psychologists’ responses. Common themes among special education teacher interviewees included their administrators and PBIS teams giving and receiving feedback on program implementation and rewards; of note, these themes were not evident in the school psychologists’ responses.

In summary, school faculty perceive differing levels of satisfaction with their schools’ PBIS-related communication strategies based on whether their schools have provided them with opportunities for PBIS-related professional development, whether their school has a team dedicated to PBIS implementation, and whether their role is that of a special education teacher or a school psychologist. The years of experience one has either in their role generally or using PBIS do not appear to moderate their perceptions of school communication related to PBIS.

### 3.3. PBIS Logistics

Overall, participants showed some agreement that they understood the logistics of PBIS with fidelity in their schools (*M* = 5.09, *SD* = 1.17). Under the category of logistics, twenty-one interview codes were initially identified and then categorized into five themes: staff buy-in, student buy-in, long-term strategies, personalized student goals, and consistency.

Table 3 shows survey results and differences based on whether participants received PBIS-related professional development from their schools. Participants who received PD from their schools showed greater understanding of PBIS logistics in their schools than those who were either unsure or had not received PD from their schools, with *F*(2, 81) = 7.68, *p* < 0.001, ηp^2^ = 0.16. As discussed in the following section, PD is an important contributor to building buy-in to program values, and interviewees here identified both faculty and student buy-in as critical in daily program functioning. Participant 4 summarized the reciprocal nature of this relationship:

“You can tell when a class is more into doing it than not into doing it. I think seeing how a teacher reacts to it will help the students react to it. So like, if you’re not into it, your kids aren’t going to be into it”.

Table 4 shows survey results and differences based on whether participants’ schools had dedicated PBIS teams. Based on their responses when asked if their school had a specified PBIS team, participants differed in their ratings of overall understanding of PBIS logistics, with *F*(2, 81) = 5.95, *p* = 0.004, ηp^2^ = 0.13. Interviewees described how PBIS teams and administrators play an important role in cultivating buy-in by reducing workload for teachers, giving faculty autonomy in meeting individual student needs, implementing parallel reward systems for staff, and using data and communication to demonstrate program success. Participant 8 described how her school’s PBIS team and administration collaborated to build faculty buy-in: “We also do a similar thing for staff. And what our reinforcer is, is the special front row parking spot. At each of our faculty meetings, our principal will have something, like a gift card or a book”.

Table 2 shows survey results and differences between special education teachers’ and school psychologists’ responses. School psychologists and special education teachers did not differ significantly in their overall ratings of logistical understanding, with *F*(1, 83) = 3.12, *p* > 0.05, ηp^2^ = 0.04. Level of experience in one’s role and level of PBIS experience were not associated with differences in overall ratings of logistics. Long-term strategies such as differential reinforcement and reward strategies based on student age and continuous, systemic framework use from year to year emerged from interviews as effective ways to implement PBIS across the entire school. Participant 5 demonstrated how this may look: “We front load the younger kids, like the kindergartners get like 500 [points] before the end of the year because they’re being taught what expected behavior looks like….I’d say in third and fourth, it’s much more intermittent”. Despite differential techniques, consistency within individual faculty members emerged as an important contributor, as illustrated by Participant 2: 

“And kids check, they look back and check. Oh, we can do that in here, or we can curse in here? Oh, I can do that! So it’s that inconsistent, that you’re not being consistent with your expectations. And that’s what, your expectations have to be high”.

### 3.4. Professional Development

When prompted, 43 participants (49.4%) reported that they had received professional development (PD) related to PBIS from their school. Overall, participants were moderately neutral regarding the efficacy of their PD experiences (*M* = 4.64, *SD* = 1.26). Within the interview category of professional development, seventeen interview codes were initially identified and then categorized into four themes: building faculty buy-in, heavy initial professional development, ongoing administrator-led development, and new staff training opportunities.

A theme of deficits in PD for new staff emerged in the interviews, with participants sharing a need for more intensive PBIS training for new staff. Participant 8′s response describes this issue:

“They don’t have the experience around the PBIS. And I think that there should be more training at their level, like when new people are coming in, whether they be new to the profession or just new to the district because every district does it a little bit differently”.

Some participants also discussed long-term deficits in PBIS-related PD over time as it transitioned from a district-wide initiative to a school-based initiative. This is evident in Participant 5′s description of initial PD opportunities compared to ongoing opportunities:

“It was at least three times that [first] year [of implementation], if not four. And I want to say it went into the second one. So after that it became implementation. And I really can’t, I cannot remember ever receiving formal PD after that”.

Not all participants experienced a deficit in PD over time. Instead, they described how administrators took the lead in carving out time to focus on PBIS throughout the year, as evidenced by Participant 7: “We have the faculty meeting every month. And sometimes that faculty meeting time is used as professional development towards the PBIS…that time is really used to kind of really go through these positive intervention plans”. 

Several interview participants, including the school psychologist, described faculty group work as a strength of their PBIS training. Participant 2 suggested this promotes a sense of cohesion among faculty and a focus on training all staff regardless of area:

“The administration will usually break us up into groups of people that usually probably wouldn’t come together from different departments and throw us all [together]. Which is kind of nice because you get the opinion of a P.E. Teacher”.

Participants also noted the importance of PD for building staff buy-in to PBIS. They discussed use of concrete applied examples, considering how PBIS can affect students, and autonomy in setting behavioral expectations during their PD.

### 3.5. PBIS and COVID-19

Overall, participants were neutral as to whether they had received the resources and tools necessary to implement PBIS effectively during the pandemic (*M* = 4.11, *SD* = 1.75). Within the category of COVID-19, twenty-two interview codes were initially identified and categorized into four themes: online learning difficulties, behavioral changes, burnout and survival mode, and strategy shifting.

Table 3 shows survey results and differences based on whether participants received PBIS-related professional development from their schools. Agreement with whether participants had received the resources and tools they needed to implement PBIS during the pandemic differed significantly based on participants’ responses to whether the school provided PD related to PBIS implementation, with *F*(2, 81) = 4.91, *p* = 0.01, ηp^2^ = 0.11. Gabriel’s post hoc comparisons showed that participants who received PD agreed more strongly that they received needed tools and resources during the pandemic (*M* = 4.59, *SD* = 1.60) when compared with participants whose schools did not provide PD (*M* = 3.32, *SD* = 1.72).

Several participants shared concerns about increased behavioral needs during online learning and the return to in-person learning, as Participant 6 explains: “Some of the behaviors have worsened because they’re not used to the structure, the demands of the classroom, whereas like, you know, all the stuff that was like on virtual was way less demanding. Let’s be honest”. Because of this, some participants discussed teachers’ lack of faith in PBIS and ultimate use of multiple reinforcement systems to immediately address behaviors rather than rely on PBIS point systems and reward earning, as mentioned by Participant 5: “I think that they’re looking at more, maybe this isn’t enough, right? You know, this isn’t shaping their behavior because it’s not immediate, right? And regular ed teachers need immediacy”.

Other PBIS-aligned methods of strategy shifting were also presented in several interviews. The most common of these was the use of in-class videos substituted for previous whole-school assemblies that teach behavioral expectations and reward students. Several participants shared how their school staff recorded skits to teach behavioral expectations to students, as shared by Participant 3:

“During the pandemic, they did it and they filmed themselves like in a Zoom type thing. So all the kids could tune in, so we still would have our [PBIS] rallies to still teach our behaviors and just have fun with the kids”.

Table 4 shows survey results and differences based on whether participants’ schools had dedicated PBIS teams. Whether the school had a PBIS team predicted participants’ level of agreement with whether they received needed PBIS support during the pandemic, with *F*(2, 81) = 5.10, *p* = 0.008, ηp^2^ = 0.11. Gabriel’s post hoc comparisons revealed that participants whose schools had a PBIS team showed greater agreement that they received needed tools and resources during the pandemic (*M* = 4.66, *SD* = 1.74) compared with participants whose schools did not have a PBIS team (*M* = 3.46, *SD* = 1.50). In the interviews, both administrator and PBIS team involvement emerged as key to continued PBIS success during COVID-19, whether it be in maintaining focus on PBIS values, using supportive messaging, creating additional visual aids, engaging classrooms on an individual level, modifying available rewards, or creating online methods of point tracking.

Table 2 shows survey results and differences between special education teachers’ and school psychologists’ responses. Special education teachers and school psychologists did not significantly differ in their level of agreement as to receiving the tools and resources they needed to implement PBIS effectively during the COVID-19 pandemic, with *F*(1, 83) = 2.15, *p* > 0.05, ηp^2^ = 0.03. Level of experience in one’s role and level of experience in a PBIS school were not associated with significant differences in ratings for whether participants received needed resources and tools to implement PBIS during the pandemic. Both new and experienced teachers described the difficulties COVID-19 presented on PBIS implementation, including challenges related to balancing curriculum demands and behavior management, hybrid learning modalities, and low student engagement. An overall sense of burnout was expressed by several interview participants due to factors including increase in demands on faculty, exhaustion, and social–emotional difficulties for both students and faculty. Although participants did not often describe a total abandonment of PBIS during the pandemic, a general sense of exhaustion seemed to permeate school functioning, as summarized by Participant 5:

“I see teachers still using the PBIS language. You know, it’s expected behavior around here. So it’s a lot of that. But they’re just weary. They’re just, I see weariness. And that’s everything, but it’s definitely with the behavior stuff”.

### 3.6. Overall Implementation

Participants showed some agreement with items related to their overall satisfaction with PBIS in their schools (*M* = 5.16, *SD* = 1.44). Within the category of overall implementation, eleven interview codes were initially identified and categorized into three themes: connecting to PBIS values, systemic factors, and connection to special education values.

Table 3 shows survey results and differences based on whether participants received PBIS-related professional development from their schools. Overall implementation ratings differed significantly based on school provision of PD (*Welch’s F*(2, 26.12) = 6.84, *p* = 0.004, ω^2^ = 0.15), such that participants who had received PD from their schools were more satisfied with PBIS implementation overall compared to participants who had not received PD. Interview participants identified buy-in to program values as an important outcome of PD for both students and faculty and here identified connection to PBIS values as a contributor to overall program success. Participant 5 described how her PBIS-related PD increased her buy-in as a special educator:

“I think the special ed teacher part of me just understood it, right, intuitively. That’s what I was doing anyway. Now you gave a name to it and you gave me a lot of formatted, you know, ideas and graphs that I could use”.

Table 4 shows survey results and differences based on whether participants’ schools had dedicated PBIS teams. Summary scores for overall implementation also differed significantly based on presence of a PBIS team (*H*(2) = 7.46, *p* = 0.02), with a mean rank score of 46.71 for those who answered yes, 33.46 for those who answered no, and 30.80 for those who were unsure. The need for school community investment was best described by Participant 8: “It takes a village, and you just need to work together at it”.

Participants’ ratings of overall implementation were not statistically significantly different based on level of experience in one’s role and level of experience in a PBIS school. The importance of faculty investment overall in promoting student outcomes was best described by Participant 1:

“I would question [teachers] if they can’t [implement PBIS]…if you don’t realize that this is essential and has to be done for [students] to be successful adults, then you really have had a cake life because 90% of their life is going to be trying things that are going to be tough. So if they think that that’s extra work, then they just don’t get it”.

This was echoed in participants’ statements regarding how critical student buy-in is for program success, including peer-to-peer support, investment in reward systems, and taking ownership of behaviors. Participant 4 detailed how faculty buy-in can encourage students to ultimately support one another in their ownership of behavioral goals:

“A good thing that my school does is the culture of getting excited for your classmate when they do something well, like showing appreciativeness to your fellow classmate for doing something that might be hard for them, like having a goal that they were able to achieve”.

## 4. Discussion

This study gives voice to educators and school psychologists at a unique time, providing an opportunity for firsthand experiences during the COVID-19 pandemic to be shared and learned from. Because the pandemic disrupted typical school functioning for an extended period of time, we aimed to understand how school faculty were perceiving and managing school-wide PBIS programs following pandemic closures. These findings demonstrate not only overall levels of faculty satisfaction, but also practical adjustments that can be made by school districts, including increasing PD and administrative support, to benefit PBIS functioning and increase school faculty satisfaction on a broader scale. Overall, participants were neutral as to whether they received necessary PBIS implementation support during the pandemic. Interview participants described several difficulties related to PBIS service provision during the pandemic, including online learning struggles, increased behavioral difficulties, necessary strategy changes, and an overall sense of burnout and going into “survival mode”. These sentiments echo a January 2022 National Education Association poll in which members rated burnout and student behavioral difficulties as serious current issues facing education [19]. COVID-19 has had a clear disruptive effect on students’ learning, forcing school staff to put time and energy into finding creative solutions, including the continuation of PBIS in virtual classrooms.

Nevertheless, interviewees shared some counterexamples of administrators working in tandem with PBIS teams to bolster PBIS implementation during COVID-19, including engaging classrooms on an individual level and creating online methods of point tracking. Overall, administrative support was revealed as a major contributor to PBIS program effectiveness, reflecting the research of Lawson et al. [24]. Interview participants’ responses also aligned with Rossi [11] in identifying provision of resources and modeling of expected behaviors as two important elements of support that administration can provide. Participants also noted administrator buy-in as an important piece of implementation tone setting [12].

Administrative support may be uniquely tied to two significant indicators of PBIS satisfaction: the provision of PD and the presence of a PBIS team. Participants whose schools provided them with PBIS-related PD indicated higher levels of satisfaction on all but one item when compared to their peers who did not receive PD. Interview participants described the intensity of their initial PD as a strength, including multiple days of PD at the beginning of a school year and opportunities for refreshers provided by administrators throughout the year during staff meetings. Faculty also described how the use of PBIS expert coaching and concrete examples increased their buy-in and understanding of PBIS, which concurs with the work of Lawson et al. [24]. These findings are notable, as PBIS implementation fidelity relies on annual faculty development and correct use of school expectations, rules, and behavioral reward systems for its success [8]. Interviews also revealed PD to be an important source of faculty buy-in to PBIS program values through modeling of positive language and opportunities to ask concrete questions about application. This aligns with the work of McKevitt and Braaksma [15], who described PD as key to ensuring that staff understand implementation expectations, that they are knowledgeable about the framework and three tiers of prevention, and that ongoing issues or concerns can be addressed in a learning environment. 

However, only 49.4% of the participants stated they had received PBIS-related PD from their school, and only 51.7% stated their school had a PBIS team, which violates expected procedures for PBIS implementation fidelity [8,14]. This suggests one potential explanation for the overall impact of both PD and PBIS teams on results, which is that they function as indicators of school investment in PBIS as a whole. This explanation implies that schools without PBIS teams or available PD are not only likely to have less satisfied school staff, but also that staff are less likely to buy in to PBIS due to lack of available knowledge of program values and benefits. This can undermine PBIS success, as faculty buy-in was described as crucial to PBIS functioning during interviews. These results convey a call to action for schools who may not have provided PD for their faculty recently or at all, as it is a key best practice for program success.

Special education teachers showed greater overall satisfaction than school psychologists on all measures of administrative support and communication practices. As previous research has suggested that special education teachers may be more confident than other school staff in their use of PBIS [21], it is not necessarily surprising that the special education teachers surveyed rated their school support and communication more positively. School psychologists also have varying responsibilities in their schools that do not involve larger group instruction or direct behavior management. Due to this, it is possible that administrators perceive them as needing less support surrounding PBIS, and that other school personnel see less importance in communicating about PBIS with them. As school psychologists can and do contribute to PBIS at a systems level by serving on PBIS teams, assessing implementation fidelity, and providing guidance for data-based decision-making practices [15,26,27], it is concerning that they may not be receiving necessary support from administrators and important PBIS-related communication. A takeaway for best practice is for administrators to ensure that school psychologists are not only included in regular PBIS communications and feedback, but also given opportunities to participate in the positive culture of PBIS.

Finally, almost no significant differences in ratings were seen based on either how much experience a participant had in their role as either a special education teacher or a school psychologist, or how much experience a participant had working in a school that uses the PBIS framework. It is notable that one’s level of experience overall did not play a substantial role in participants’ satisfaction with PBIS implementation in their schools. Rather, systemic factors, such as the existence of PD opportunities and faculty teams dedicated to PBIS implementation, as well as the dynamics of support and communication as they relate to different faculty roles, appear to be more prominent predictors of faculty satisfaction. Addressing systemic gaps and encountering roadblocks outside of one’s control can involve a significant increase in workload. However, these are modifiable factors within a school’s use of the PBIS framework when compared to long-term increases in faculty knowledge and experience over time. These can be seen as overall positive findings for schools looking to increase faculty satisfaction and fidelity of implementation related to PBIS.

## 5. Limitations and Future Directions

Several limitations emerged in this study that may be addressed in future research. First, only one school psychologist volunteered to be interviewed. This made it difficult to determine what unique factors may be contributing to their experience, as they showed significant differences in their ratings of administrative support and school communication practices when compared to special education teachers. This may be indicative of the nature of a school psychologist’s everyday demands, as the heavy workload of a practicing school psychologist may prevent them from engaging in “extra” experiences, such as participating in psychological research. Future research could seek to recruit primarily school psychologists to gain a fuller picture of their experience through interviews and survey measures.

Additionally, the participants of this study were largely white and female. Although this is not a representative sample of the U.S. population more broadly, this sample is representative based on the current demographic makeup of the school psychology and special education fields. Future research may seek to recruit school psychologists and special education teachers from primarily racially, ethnically, or gender-diverse backgrounds to better understand their unique experiences in the field of education.

Based on the results of the current study, future research should aim to engage school administrators in their perceptions of PBIS effectiveness within their schools, as administrative practices were seen to be a key asset to school faculty satisfaction in this study. While administrators provide leadership at the school level, they also must navigate district-wide expectations that pose unique challenges and requirements regarding school-wide programming efforts. Using interviews to gain a better understanding of the PBIS strategies administrators use and the challenges they face in implementation may aid in discovery of best practices that are feasible for all school stakeholders.

## Figures and Tables

**Table 1 behavsci-13-00372-t001:** Participant demographics related to racial–ethnic identity, gender, education, location, school type, and grades served.

Characteristic	Full Sample		School Psychologists		Special Education Teachers		Interview Participants	
	*n*	%	*n*	%	*n*	%	*n*	%
Total	87	100	29	33.3	58	66.7	8	9.2
**Racial-Ethnic Identity**								
White/Caucasian	67	77.0	21	72.4	46	79.3	7	87.5
Hispanic/Latino	5	5.7	3	10.3	2	3.4	0	0
African American/Black	5	5.7	1	3.4	4	6.9	0	0
Biracial	2	2.3	1	3.4	1	1.7	0	0
Asian	1	1.1	0	0	1	1.7	1	12.5
Middle Eastern	1	1.1	0	0	1	1.7	0	0
**Gender**								
Female	71	81.6	28	96.6	43	74.1	8	100
Male	14	16.1	0	0	14	24.1	0	0
**Highest Level of Education**								
Bachelor’s degree	7	8.0	0	0	7	12.1	0	0
Some graduate school	4	4.6	0	0	4	6.9	1	12.5
Master’s degree	53	60.9	9	31.0	44	75.9	5	62.5
Specialist degree	15	17.2	13	44.8	2	3.4	2	25.0
Doctoral degree	8	9.2	7	24.1	1	1.7	0	0
**U.S. State of Practice**								
New York	37	42.5	12	41.4	25	43.1	3	37.5
New Jersey	18	20.7	8	27.6	10	17.2	2	25.0
Connecticut	15	17.2	5	17.2	10	17.2	2	25.0
Nebraska	8	9.2	0	0	8	13.8	1	12.5
Other	6	6.6	2	6.8	4	4.8	0	0
**Type of School**								
Public	79	90.8	29	100	50	86.2	7	87.5
Charter	3	3.4	0	0	3	5.2	0	0
Private	4	4.6	0	0	4	6.9	0	0
Board of Cooperative Educational Services (BOCES) District	1	1.1	0	0	1	1.7	1	12.5
**Number of Grade Levels Served**								
One to Three	49	56.3	5	17.2	44	75.9	7	87.5
Four to Six	23	26.3	11	37.9	12	20.6	1	12.5
Seven to Nine	9	10.3	8	27.6	1	1.7	0	0
Ten or more	6	6.7	5	17.1	1	1.7	0	0

**Table 2 behavsci-13-00372-t002:** Survey results based on role in school.

Measure	Special Education Teachers		School Psychologists		*F*	*p*	ηp^2^
	*M*	*SD*	*M*	*SD*			
Administrative Support Subscale Summary Score	5.08	1.25	4.12	1.65	(1, 81) = 8.59 **	0.004	0.10
Communication Practices Subscale Summary Score	4.61	1.60	3.55	1.60	(1, 83) = 8.18 **	0.005	0.09
PBIS Logistics Subscale Summary Score	5.26	1.10	4.76	1.26	(1, 83) = 3.12	>0.05	0.04
I received the tools and resources that I needed to implement PBIS effectively during the COVID-19 pandemic.	4.31	1.60	3.72	1.98	(1, 83) = 2.15	>0.05	0.03
Overall Implementation Subscale Summary Score	5.28	1.26	4.91	1.73	Welch’s (1, 40.70) = 0.96	>0.05	ω^2^ = 0.01

Note: Welch’s *F* tests were used when homogeneity of variance assumption was violated. ** = significant at 0.01.

**Table 3 behavsci-13-00372-t003:** Survey results based on school provision of professional development.

Measure	Yes		No		Unsure		*F*	*p*	ηp^2^
	*M*	*SD*	*M*	*SD*	*M*	*SD*			
Administrative Support Subscale Summary Score	5.45	1.14	3.85	1.39	4.41	1.49	(2, 80) = 13.07 ***	<0.001	0.25
Communication Practices Subscale Summary Score	5.05	1.45	3.29	1.49	3.85	1.51	(2, 82) = 12.63 ***	<0.001	0.24
PBIS Logistics Subscale Summary Score	5.56	1.46	4.46	1.44	4.71	1.11	(2, 81) = 7.68 ***	<0.001	0.16
I received the tools and resources that I needed to implement PBIS effectively during the COVID-19 pandemic.	4.59	1.60	3.32	1.72	4.31	1.75	(2, 81) = 4.91 **	0.01	0.11
Overall Implementation Subscale Summary Score	5.74	0.81	4.42	1.83	4.97	1.41	Welch’s (2, 26.12) = 6.84 **	0.004	ω^2^ = 0.15

Note. Welch’s *F* tests were used when homogeneity of variance assumption was violated. ** = significant at 0.01 *** = significant at 0.001.

**Table 4 behavsci-13-00372-t004:** Survey results based on response to PBIS team criterion.

Measure	Yes		No		Unsure		*F*	*p*	ηp^2^
	*M*	*SD*	*M*	*SD*	*M*	*SD*			
Administrative Support Subscale Summary Score	5.12	1.26	4.46	1.66	4.02	1.33	(2, 81) = 3.40 *	0.04	0.08
Communication Practices Subscale Summary Score	4.69	1.44	3.78	1.82	3.80	1.82	(2, 83) = 3.31 *	0.04	0.08
PBIS Team Members Subscale Summary Score (*n* = 15)	5.81	0.87							
PBIS Team Non-Members Subscale Summary Score (*n* = 28)	4.90	1.50							
PBIS Logistics Subscale Summary Score	5.51	1.15	4.65	1.09	4.68	0.97	(2, 81) = 5.95 **	0.004	0.13
I received the tools and resources that I needed to implement PBIS effectively during the COVID-19 pandemic.	4.66	1.74	3.46	1.50	3.60	1.58	(2, 81) = 5.10 **	0.008	0.11
Overall Implementation Subscale Summary Score	5.55	1.26	4.73	1.60	4.73	1.30	*H*(2) = 7.46 *	0.02	

Note: Kruskal–Wallis *H* tests were used when normality assumption was violated. * = significant at 0.05; ** = significant at 0.01.

## Data Availability

Data are not publicly available due to privacy and ethical restrictions specified in our IRB document. However, if researchers are interested in obtaining the related data, they can reach out to the corresponding author at hterrell@fordham.edu for individual use.

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
