# Peer review of "Translating School Faculty Experiences Using PBIS into Recommendations for Practice"

_behavsci, 2023, doi:10.3390/bs13050372_

Round 1

Reviewer 1 Report

The introduction presents the subject well, is very well documented and clearly structured. The whole is very complete. However, I would have appreciated if the second paragraph had been more documented/supported by other references supporting the importance of the PBIS framework (and not just by one reference).
As well, subsection 1.5 should in my opinion be more detailed and referenced since this section clearly supports the objectives of the article (and particularly the first research question).

In terms of methodological aspects, the 2nd research question is closed (it can be answered with a yes or no) and would deserve a more open formulation, especially because of the method (MMR) used.

Concerning the interview participants, it would be relevant to explain further how only 8 people participated, while 36 gave their consent to be interviewed. What were the selection/retention criteria? At this stage of the article, we also do not understand why only 1 psychologist was selected for the interviews. Why wait until the "limitations" section to have the explanation?

It would also be more appropriate to use McDonald's omega index rather than Cronbach's alpha to assess the reliability of the measurement instrument (see e.g., Goodboy & Martin, 2020, https://doi.org/10.1080/23808985.2020.1846135 or Béland et al, 2017, https://doi.org/10.7202/1050915ar).

In addition, I am not sure that the authors are performing Grounded Theory. Indeed, what is produced in this article hardly meets Paillé's (1996) definition: a method of data analysis "aimed at inductively generating a theorization about a cultural, social, or psychological phenomenon, by proceeding to the progressive and valid conceptualization and linking of qualitative empirical data" (p. 184). What is missing is the more obvious or explicit linking of data and, most importantly, the integration, modeling, and theorizing steps. Integration consists in bringing out the problematic, which is never posed a priori. At this point in the analysis of field data, the researcher asks himself what the problematic of his field investigation is. This is not the case in this article. Modeling consists of producing a theoretical diagram that reveals the dynamics of the phenomenon under study. We do not find this step in the submitted text. Finally, theorizing is the stage where the researcher puts into form, based on the preceding stages, the theoretical implications of his research work.

In my opinion, in this text, only the first two steps of grounded theorizing are used to analyze the data (e.g., coding and categorization), but the following steps are missing to claim grounded theorizing. It seems to me that, in this text, we are more in a thematic content analysis as we see in some phenomenological studies. The results do not really lead to a theory from the qualitative data.

At the formal level, the abbreviation "BOCES" means nothing to a non-American reader (p. 6, Table 1). It would be relevant to make it explicit. In addition, the title of section 3.5 is not italicized.

Reviewer 2 Report

April 14th, 2023 

Review: Translating School Faculty Experiences Using PBIS Into Recommendations for Practice

Thank you for the opportunity to review article titled Translating School Faculty Experiences Using PBIS Into Recommendations for Practice.

Below are some points of constructive critique and chances to improve this manuscript.

Abstract

It is a bit strange that the author(s) highlight the burnout of special education teachers and psychologists because the research is not about burnout but their experiences on using PBIS.

Also, pointing “the reason for this remains unclear” needs to be replaced by the results that are clearer based on this study.

Introduction

Page 2: Correct McKevitt & Braaksma to McKevitt and Braaksma.

Method

This study is an explanatory sequential mixed-methods design, whereby the results of an initial quantitative segment are further elucidated in a secondary qualitative. Please, explicate how seven special education teachers and one school psychologist can, and did, elucidate quantitative results. Also, clarify what are “unwanted order effects”?

“ANOVA F tests were used to determine group mean differences related to the…” is a replicate of what was already said on page 5.

Welch's F test needs more detailed description. The same is true with Gabriel’s post-hoc test, Kruskal-Wallis H, and Mann-Whitney U. Why are they correct methods for this study?

ηp2: Check APA style for notation.

Results

Table 2: Note. Welch’s F tests and Mann-Whitney U tests were used when homogeneity of variance assumption was violated. Kruskal-Wallis H tests were used when normality assumption was violated. Yes, but where are the results when they were used? I don’t see F*, U, or H values in the table. The same is true with other tables.

Discussion

The authors say that “the pandemic has necessitated new research on how school faculty are perceiving and managing school wide PBIS programs following pandemic closures, a gap that is filled by this study”. Be specific why has pandemic necessitated this kind of study.

I wouldn’t say that “students of color made up 54% of all public school students in Fall 2020, showing a wide discrepancy between service providers and the students they work with. Efforts to diversify the field are more necessary than ever considering the demographic makeup of students served by this largely white and female population” is a limitation of this study.

As reviewed above, I have some concerns. Therefore, I recommend revising this manuscript. Hope my comments help!

Round 2

Reviewer 2 Report

Accept in present form.